# Ain’t Too Proud to Beg! Effects of Leader’s Use of Pride on Groups

**DOI:** 10.3390/ijerph17197146

**Published:** 2020-09-29

**Authors:** Catherine S. Daus, Stephen R. Baumgartner

**Affiliations:** Department of Psychology, School of Education and Health and Human Behavior, Southern Illinois University Edwardsville, Edwardsville, IL 62026-112, USA; stbaumg@siue.edu

**Keywords:** leadership, pride, authentic pride, hubristic pride, task satisfaction, group cohesion, leader satisfaction

## Abstract

Studies of discrete pride in the workplace are both few and on the rise. We examined what has, to date, been unstudied, namely the impact that a leader’s expressions of authentic and hubristic pride can have on the followers at that moment, and on their attitudes regarding their task, leader, and group. Students working in groups building Lego structures rated their perceived leader regarding expressions of pride, both authentic and hubristic. Students who perceived the leader as expressing more authentic pride rated the task, group (satisfaction and cohesion), and leader more positively, while the reverse was generally true for perceptions of expressions of hubristic pride. We found these effects both at the individual level and at the group level. We also predicted and found moderation for the type of task worked on, creative or detailed. Implications abound for leader emotional labor and emotion management.

## 1. Introduction

“Pride is as loud a beggar as want and a great deal more saucy.”—Benjamin Franklin [1].

As the above quote from Ben Franklin alludes to, pride can be a powerful feeling and motivator, in one’s work life, or just generally. Housed in the category of the “self-conscious emotions” [2], historically, pride has been (and often is still) viewed negatively [2,3,4], as it seems to imply, and indeed reflects, some self-centeredness and even selfishness [5]. In fact, in a review of humility, Tangney [6] (p. 412) cited a 1998 Oxford English Dictionary definition which contrasted humility with “pride or haughtiness,” thus equating the two.

Yet, like Tangney [6] claimed regarding humility, pride is much more complex of an emotion. Specifically, pride has two recognized dimensions of authentic and hubristic pride, which represent differing foci and expressions (see Schaumberg & Tracey [7] for a recent review). Regarding expressions of pride, to simplify somewhat, authentic pride, or pride in one’s accomplishments, is seen as legitimate and is respected, while hubristic pride, or pride generally in oneself, is viewed much less positively.

When we imagine the influence that one’s expressions of emotions potentially have on others, perhaps nowhere is more compelling a place to focus than on leaders. It is well documented that leader emotional expressions and management have important effects on followers/employees (see Kaplan, et al. [8], for a theoretical model and review of literature of leader—employee emotion management). These effects include influences on performance, turnover/withdrawal intentions, psychological well-being, and a host of work attitudes, such as job and leader satisfaction. In a recent review, Schaumberg and Tracy [7] (p. 16) discuss that, while much evidence supports that both authentic and hubristic pride promote leadership emergence likely through different routes, they make the following observation: “Whether or under which circumstances authentic pride and hubristic pride relate to leadership effectiveness remains a more open question”. Our research seeks to contribute towards answering this question. We examine the effects of perceived expressed pride of leaders on the core critical group attitudes of leader, group, and task satisfaction. Understanding how leaders’ emotions expressions are perceived and how they are then related to group attitudes has the potential to enlighten both organizational leader theory and application, with the possibility of helping leaders to understand that perceptions of their group members can matter greatly in the overall health and well-being of the group. Below, we outline the specific aspects of leader emotion (pride) and predicted outcomes we studied, with their theoretical and empirical undergirding.

### 1.1. Review of Literature

#### 1.1.1. Examining Pride

While much research has focused on the discrete (primary) emotions of happiness, sadness, fear, surprise, anger, and disgust [9], pride has not received as much scholarly attention [7]. Pride has previously been referred to as being a part of the secondary class of emotions, distinct from basic emotions, such as happiness and sadness [10], yet recent research has suggested that pride meets criteria to be considered a basic emotion, as it has a unique nonverbal expression that is recognized cross-culturally [9,11]. This “rise in status” of pride highlights its increasingly recognized importance.

Pride is an emotion that arises from both satisfaction with one’s achievements and successes (i.e., authentic pride), or a positive globalized vision of oneself (i.e., hubristic pride) [2,7]. Authentic pride might therefore be characterized as “I did a good thing” while hubristic pride might be characterized as “I am good.” Pride is indicated as very influential in motivation, as it helps regulate and maintain self-esteem [9]. Also, this influence on self-esteem aids in reinforcing prosocial behaviors, such as achievement and caregiving. Pride’s boost to self-esteem may indicate to the individual that their behaviors are socially valued, thereby reinforcing those behaviors [7].

Pride also plays a role in social hierarchies and can be used to express strength, achievement, value, and gain in social ranking [7,12]. Feelings of pride can serve as a social marker, indicating to the individual his/her rank in society [13]. Pride plays an integral part in many domains of human functioning, including group dynamics. Given this, the distinction between authentic and hubristic pride warrants further investigation as these both appear to elicit different kinds of responses from individuals perceiving the authentically or hubristically proud individual.

#### 1.1.2. Authentic Versus Hubristic Pride

Pride has been explained as comprising two distinct facets: authentic pride and hubristic pride [9,14]. Authentic pride is related to specific achievements, self-esteem, agreeableness, and conscientiousness [7,14]. Williams and Desteno [13] suggest authentic pride stems from a particular event or situation and is usually based on specific accomplishments and paired with authentic feelings of self-worth. Authentic pride may also serve as an indicator of the prosocial methods the individual uses to attain social status, especially when compared with hubristic pride [15].

The authentically proud individual is more likely than the hubristically proud individual to define success as the mastery of a skill or completion of a task. Additionally, it has been found that individuals who tend to attribute a range of events more to effort also tend to respond to events with authentic pride, whereas those who tend to attribute such events more to ability are more likely to respond with hubristic pride [7,14]. In sum, authentic pride is considered a prosocial emotion because it fosters personal wellbeing, as well as the acquirement of skills, genuine self-esteem, and perseverance at difficult tasks [15].

In contrast, hubristic pride is tied more to attributions of personal excellence than to one’s accomplishments. A hubristically proud individual often seeks status enhancement, dominance, and admiration. Also, those with hubristic pride exhibit uncaring exploitive behaviors towards others [16,17]. In a study by Carver et al. [18], hubristic pride was also related to impulsivity and aggression, while authentically proud individuals were found to display more self-control. Given these findings, the hubristically proud employee may be seen as narcissistic, egotistical, arrogant, and boastful, viewing success only as self-accomplishments, and paying little or no attention to the group.

Taken altogether, we see that authentic pride is associated with positively valanced constructs, such as self-worth/self-esteem, achievement, goal engagement, prosocial behavior (intentions and actions), positive social relationships, support, and empathy for others, and mental health. In contrast, hubristic pride is associated with negative constructs, such as arrogance, narcissism, need for public recognition, social dominance, prejudice, anger, and aggression [2,7,13,15,18,19,20,21]. Next, we discuss how pride may have important organizational consequences. We first discuss research broadly across outcomes, and then focus on our three chosen outcome variables of satisfaction with leader, task satisfaction, and group cohesion/satisfaction.

#### 1.1.3. How Perceptions of Pride Impact Organizational Outcomes

We note, generally, that logic for all three of our dependent variables rests partly on the body of evidence which shows that authentic pride is more of a prosocial emotion than hubristic pride [15]. Tracy and colleagues [20] (p. 196) summarized their findings on the relationship between authentic pride and prosocial and positive outcomes with the following statement: 

“Specifically, findings demonstrate that when narcissistic and genuine self-esteem are empirically distinguished, genuine self-esteem (along with authentic pride) is positively related to successful social relationships and mental health, whereas narcissistic self-aggrandizement (along with hubristic pride) is positively related to aggression and other antisocial behaviors.”

Given this body of research in support of authentic pride as an antecedent to positive and prosocial outcomes, we expect positive associations and attitudes with perceived leader use of authentic pride compared to hubristic pride. In the following sections, we briefly discuss leadership and its general relationship to pride and follow this with an examination of our outcomes of interest, namely follower level satisfaction with the leader, satisfaction with a task, and group cohesion and satisfaction.

How leaders use their power is related to the kind of pride perceived by followers. Yukl [22] (p. 256) discusses how successful leaders should use power subtly, minimizing status differences, and making sure to affirm employees’ self-esteem. Contrasting with this is the use of power “in an arrogant, manipulative, domineering manner”. These two styles map onto authentic and hubristic pride nicely. However, very few studies have examined these constructs in relation to their impact on effective leadership.

In one of the few examples of such research, Yeung and Shen [21] examined leaders’ experiences of authentic and hubristic pride and their resulting leadership behaviors across three studies, and largely supported the above suggestions by Yukl [22]. Leaders’ trait and discrete experienced authentic pride were associated with the actual use of, and intention to use, more effective and fewer ineffective leadership behaviors. Also consistent with Yukl’s [22] findings, hubristic pride was associated with more abusive behaviors.

In a related study, Ritzenhöfer and colleagues examined leader expressions of pride and the effects on followers [5]. However, in their study, they collapsed expressions and verbal descriptions of hubristic and authentic pride labeling the combination “self-referential” pride. They found that leaders’ gratitude expressions showed a positive effect on followers, and leaders’ pride expressions that were ascribed as more selfish by followers resulted in lower satisfaction with the leader. Also, leaders’ expressions of gratitude were positively associated with subordinates’ job satisfaction, and leaders’ expressions of pride were positively associated with subordinates’ intentions to leave the organization.

#### 1.1.4. Satisfaction with Leader

Ritzenhöfer and colleagues provide the most direct logic and empirical evidence which supports our dependent variable, satisfaction with the leader [5]. Ritzenhöfer and colleagues’ study, discussed earlier, regarding gratitude supported linkages with both leader satisfaction and job satisfaction, showing that leader pride was negatively related to satisfaction with said leader, while gratitude (conceptualized somewhat as opposite to pride) was positively related to subordinate job satisfaction. As well, Spraggon and Bodolica [23] cite how authentic pride is linked to trust. They discuss how these two constructs, when considered together, explain employee satisfaction with the dyadic relationship with their supervisor.

In another paper examining governance at high levels in organizations, Bodolica and Spraggon [3] argued that authentic pride of those in upper levels of governance in organizations should be related to a host of positive outcomes, performance, and perceptions of effective leadership: “We have shown above the extent to which authentic pride can improve the ability of the representatives of higher echelons in organizations to assume successful leadership and governance roles” [3] (p. 546). One aspect of leader effectiveness is the satisfaction of subordinates. Additionally, the authors encouraged leaders to practice and learn how to develop employees’ feelings of authentic pride, suggesting it should be related to their job satisfaction.

As well, the Yeung and Shen [21] article showed that authentic pride was related to consideration (i.e., people-oriented behaviors) and initiating structure behaviors (i.e., task-oriented behaviors), while hubristic pride was associated with abusive leadership [24]. These behaviors (task and relational) should be related to positive perceptions of the leader, resulting in more satisfaction with the leader.

#### 1.1.5. Task Satisfaction

The positive relationship between certain aspects or styles of leadership and their relationship with job satisfaction has a long-standing history, and has been demonstrated in multiple contexts and with multiple dimensions of leadership. For example, recent meta-analyses confirm the importance of leader emotional intelligence on subordinates’ satisfaction [25], leader behaviors (consideration and initiating structure) on the job satisfaction of nurse faculty [26], and ethical leadership on job satisfaction [27]. Two articles reviewed above [3,5] suggested that employee job satisfaction is or should also be an outcome from authentic pride of leaders. While not the same thing, task satisfaction has long been considered one of the primary sub-dimensions of job satisfaction [28] and should thus be an expected outcome as well, perhaps even more so than job satisfaction because it is more targeted and directly influenceable by a leader who should be able to provide resources and task expertise and feedback.

In an indirect link, one line of research found that authentically proud leaders behave more ethically than hubristically proud leaders [29]. Employees whose leaders behave more ethically tend to be more satisfied with their jobs [30].

#### 1.1.6. Group Cohesion and Satisfaction

Group cohesion is defined by Carron and Brawley [31] (p. 94) as “a dynamic process that is reflected in the tendency for a group to stick together and remain united in the pursuit of its instrumental objectives and/or for the satisfaction of member affective needs.” Group satisfaction, alternatively, is best understood as a global positive assessment stemming from making progress toward the group’s goals [32]. Although both have an emphasis on instrumental (task) progress, they appear to be distinct. Back in 1965, writing on the relationship between the two, Hagstrom and Selvin [33] argued convincingly that group cohesion (“sociometric cohesion”) and group satisfaction (“social satisfaction”) are definitively separate constructs; yet they did acknowledge that there may be times when combining the two is appropriate (It should be noted that these researchers were examining friend groups). They allowed for contexts where collapsing/averaging made sense. A more current study examined how feelings of group cohesion actually predict satisfaction with the team [34], and supported a predictive model. Yet, although the two constructs are distinct, they are also highly related, as both of these articles show.

In full transparency, our original intent was to examine these constructs/variables as separate dependent variables. However, for the purpose of this study, the two constructs were combined after initial results showed a high degree of overlap. We provide psychometric justification (see methods section) for collapsing the two. In addition, we performed all analyses with the variables separated as well as combined. For parsimony, in the body of the text, we report results for the combined variable. We include any results which differ notably when analyses were run separately.

Authentic pride in a group or team or group leader is preceded by an achievement of some sort, and if that leader has the personality traits of being agreeable, conscientious, emotionally stable, and has high self-esteem. Leaders who experience authentic pride tend to attribute success to effort rather than ability or some uncontrollable attribute. Often, the feeling of authentic pride is followed by behaviors such as advice-giving, skill-sharing, and other altruistic behavior. Given this, leaders who display authentic pride are more likely to encourage prosocial behaviors and show empathy for others, as well as a willingness to collaborate with others and show concern for how their behavior impacts others [35]. In a study conducted by Hardy and Van Vugt [36], they found that individuals who engaged in altruistic actions felt authentically proud and consequently gained the acceptance and respect of the group, reflected in satisfaction with the group and feelings of group cohesion.

Above we discussed how authentic pride was related to prosocial behavior and expressions of gratitude [5,37]. In a group context, particularly, such prosocial, helpful behavior on the part of the leader should be associated with leader perceptions (as we argued above), and with general positive feelings about the group.

### 1.2. The Moderating Role of Task Type on Outcomes

Leadership occurs within specific contexts, and task type is an integral component of leadership style match and success. For example, the path-goal theory of leadership [38], proposed over 40 years ago, acknowledged the importance of task environment in predicting which leadership style would be most effective. Yet, little research has examined task type in conjunction with authentic and hubristic pride with a notable exception: Damian and Robins [39] found that those high in dispositional authentic pride performed particularly well on a creativity task, while those high in dispositional hubristic pride did poorly. We expand on their study and examine leaders’ expressions of pride.

Further supporting the importance of the interaction of leadership with task type, in a meta-analytic review of leadership behaviors and innovation, Rosing et al. [40] discussed the use of two seemingly opposing styles of leadership and their relationship with innovation. They discussed how “opening” leadership styles, characterized by fostering exploration, in contrast to “closing” leadership styles which include setting specific guidelines, monitoring progress, and taking corrective action, are alternately needed for task success. In particular, opening styles foster innovation and creativity, and closing styles foster implementation. Thus, authentic pride displays which signal openness and authenticity, should foster performance on creative tasks especially, while hubristic displays should hinder it. We thus chose those two distinct types of tasks (creative and detailed) and proposed those as moderators.

In sum, we aim to add to the literature by examining differential prediction of authentic and hubristic pride expressed by the leader on group outcomes. We expect hubristic pride to have uniformly negative relationships with attitudes of satisfaction with the leader, task, and group, while authentic pride should be positively related to them. Based on the literature presented, the following hypotheses are proposed:

**Hypothesis** **1** **(H1):**
*Group members’ ratings of leaders’ displayed authentic pride of their perceived leader will be positively associated with (a) satisfaction with leader, (b) task satisfaction, and (c) group cohesion/satisfaction.*


**Hypothesis** **2** **(H2):**
*Group members’ ratings of leaders’ displayed hubristic pride of their perceived leader will be negatively associated with (a) satisfaction with leader, (b) task satisfaction, and (c) group cohesion/satisfaction.*


**Hypothesis** **3** **(H3):**
*Hypothesis 1a–c will be moderated by task type such that the positive relationship will be stronger for those performing a creative versus a detailed task.*


**Hypothesis** **4** **(H4):**
*Hypothesis 2a–c will be moderated by task type such that the negative relationship will be weaker for those performing a detailed versus a creative task.*


## 2. Materials and Methods

### 2.1. Participants

All subjects gave their informed consent for inclusion before they participated in the study. The study was conducted in accordance with the Declaration of Helsinki, and the protocol was approved by the Ethics Committee of Southern Illinois University Edwardsville (IRB # 15-1001-2). Page: 6.

Participants included 179 introductory psychology undergraduate students at a medium-sized, Midwestern university who earned partial course credit for participating. Students in this course have a research requirement option and must participate in research studies (or complete quizzes over research articles). Students are able to choose from a variety of studies offered throughout the semester. Students who chose this study signed up for a “study of group dynamics,” and showed up to the lab at the appointed time. The study could only be run if at least three students showed up to participate. These students comprised 43 different groups ranging in size from 3–6 members. Groups were compiled based on when the person signed up for the study such that there were 3–5 slots open per experiment time. If fewer than 3 signed up, the people were notified and asked to participate in a different time. Average age ranged from 17–28; and the mean was 19.50 (*SD* = 1.92). Females predominated with 70% of the sample. Participant racial demographics consisted of 57% White, 35% Black, 3% each Hispanic or Asian, with the remainder identifying as American Indian or other.

### 2.2. Measures

#### 2.2.1. Demographics

Participants were asked to indicate their age, race, and gender.

#### 2.2.2. Task Satisfaction

The job descriptive index (JDI) [27] was adapted and utilized to measure task satisfaction. The JDI consists of 18 adjective checklist items (e.g., “fascinating,” “satisfying,” “frustrating”) which participants answer with a “yes,” “no,” or “?” if they were unsure whether the task fit the adjective description. The JDI has been found to be a highly reliable measure [41]. Coefficient alphas have been reported as high as 0.90 for the work subscale which we modified to be task-based, replacing the word “job” with the words “group Lego task,” and in our study the coefficient alpha was 0.70.

#### 2.2.3. Group Cohesion and Team Satisfaction

A measure developed by Lee and Farh [42] was utilized to determine level of group cohesion. Their measure consists of 7-items, each rated on a 7-point Likert scale anchored with 1 (*strongly disagree*) to 7 (*strongly agree*). A sample item was: “My group members helped each other on the task.” Lee and Farh [42] reported a coefficient alpha of 0.92.

To measure the participants’ sense of team satisfaction, Larson, Larson, and LaFasto’s [43] teamwork excellence measure (TEM) was utilized. The TEM consists of 7-items with a Likert scale gauging the level at which the participant felt satisfied with their team’s performance. The scale used anchors of 0 (*not at all*) to 4 (*extremely*). A sample item was: “Our team exerted pressure on itself to perform higher.” Kolb [44] used the measure and reported a coefficient alpha of 0.91.

Ultimately, through an exploratory factor analysis and examination of the scree plot, these measures were combined into one during analysis as they appeared to be measuring one unified construct of team/group-level cohesion and satisfaction. All items were thus combined and presented a Cronbach’s alpha of 0.87.

#### 2.2.4. Leader Satisfaction

Six items comprised the leader satisfaction measure, which also used a portion adapted from Larson, Larson, and LaFasto’s [43] TEM. Items were anchored for team satisfaction, with 0 ranking (not at all) and 4 ranking (extremely). A sample item was: “Our leader created a safe climate for the team’s success.” A Cronbach’s alpha of 0.84 was found in the present study.

#### 2.2.5. Leader Authentic and Hubristic Pride

A 20-item measure developed by Tracy and Robbins [13] was utilized to determine whether or not the perceived leader was seen as conveying authentic and/or hubristic pride. This measure asked participants to report perceived state authentic and hubristic pride from the perceived leader. Participants were instructed, “KEEPING YOUR IDENTIFIED LEADER IN MIND, rate them on the following items.” The authentic pride items included statements such as, “They seemed to feel accomplished.” The hubristic pride items included statements such as, “They seemed to feel arrogant.” These 20 items were rated on 5-point Likert-type scale (0 = Not at All, 4 = Extremely). The pride scales used in our study had high reliability, with a Cronbach’s alpha of 0.88 for authentic pride, and 0.90 for hubristic pride.

#### 2.2.6. Leadership Style

Originally, we planned to examine leaders’ own self-report of their leadership style [45] as a potential control variable, and thus participants filled out a leadership styles self-report measure, but due to poor reliability of this measure, we could not use it for any analyses.

### 2.3. Procedure

Initially this study was designed to be fully experimental, with a confederate leader who acted in either an authentic or hubristic prideful way. After extensive piloting of scripts (to be either authentic or hubristic) and employing acting students to be the confederate, we discovered this method not to be feasible. This was because of the necessary improvisational nature of responses from confederates to changing events and comments from participants, which did not allow for psychological fidelity [46], meaning the scripts simply could not be held constant (experimentally) and still represent a realistic situation. As responding in a scripted manner would likely reveal the identity of the confederate actor, this original method was changed in favor of a more ecologically valid design. Instead of introducing a confederate as a leader, participants rated one another on who they perceived to be the leader. Participants were then asked (at the end of the experiment) to rate that person’s expressions of authentic or hubristic pride.

Participants signed up for a group and leadership study through the psychology department’s participant pool which utilizes students in introductory psychology and gives research credit experience to them for participating in studies. In this study, it was required that three students sign up and attend their selected research session in order for the session to be conducted. Participants were assigned to a condition/session that had been randomly determined to be the “creative” or “detailed” Lego construction task. When participants arrived, they were instructed to wear a name tag that would be visible during the study. This was done as to make identification of a perceived leader easier when asked to do so later in the study. In both tasks, participants had a chance to win a prize of a $15 gift card to Jimmy John’s, a national fast food sandwich chain restaurant, which was shown to participants before the researchers read the direction prompt. For the creative task, participants were told:

“…you will be building a creative Lego structure as a group. The group is responsible for coming up with ideas. As I had already mentioned, there will be a chance to win a prize. Across all groups that participate in the study, the most creative group will win a prize. First place participants will each receive a $15 gift card to Jimmy John’s (motion to actual gift cards), and second place participants will each receive a $5 gift card to Jimmy John’s.”

For the detailed task, participants were shown a model structure of either a helicopter, a house, or a bird, which was already built, and told:

“…you will be creating a detailed Lego structure as a group. As I mentioned, there will be a chance to win a prize. Across all groups that participate in the study, the group that most closely matches the model will win a prize. If multiple groups match the model exactly, the group that completed it the fastest will receive the prize. First place participants will each receive a $15 gift card to Jimmy John’s (motion to actual gift cards), and second place participants will each receive a $5 gift card to Jimmy John’s.”

Participants filled out measures of demographics and personal leadership styles (originally intended for use to corroborate the cover story for how a leader was chosen but intended to be explored as a predictor and control variable in our analyses). They were then given 30 min to build their structure. After the 30 min had elapsed or if groups indicated they were finished before 30 min, the time taken to complete the structure was recorded, and a picture was taken of their structure. This picture was planned to be rated later for either accuracy or creativity (Records of some groups’ “performance” were lost; therefore, we could not analyze this data). After task completion, participants filled out five measures: task satisfaction; team cohesion/satisfaction; leader satisfaction; leader authentic/hubristic pride display (Measure regarding the leader satisfaction and authentic/hubristic pride displays were introduced with the following prompt: “Who do you believe exerted the most leadership behaviors/qualities?).

### 2.4. Data Analysis

Analyses were conducted at both the individual level and the group level. The primary analysis at the individual level utilized each person’s perceptions of whomever they thought the leader was. (See above measures section).

For the group level analysis, in order to justify this statistically, group members had to have “enough” agreement about three (broad) things: (1) who the leader was of the group; (2) assessments of said leader regarding perceptions of authentic and hubristic pride expressed; and (3) the dependent variables. For the first “cut” of agreement as to whom the leader was, some groups had such diverging opinions as to whom the leader was, there was no way to collapse regarding perceptions of leader in the group, because they were all rating different leaders in their mind. This process (described below) left a total sample of *N* = 22 groups. Because there were five more variables to assess agreement on (two IVs and three DVs), no single group had strong enough agreement on all five variables to enable collapsing. We therefore ran group analyses on the 22 groups.

For both levels of analyses, we first checked normality assumptions and corrected when necessary. Then, we ran one-way Pearson correlations, followed by hierarchical regressions, including demographic control variables if they showed significance in the correlations. If the control variables were non-significant in the regression analyses, we re-ran the regressions with those variables removed, for parsimony and present the skimmed regression results. Finally, for the moderation analysis, we utilized Andrew Haye’s PROCESS macro add-in for the Statistical Package for the Social Sciences (SPSS) software [47].

#### 2.4.1. Group Data Aggregation

Data for group-level variables were compiled by examining if the group attained majority member agreement (i.e., 66%) on who the perceived leader of the group was. This was done by tallying votes on who the members of the group believed the leader of the group was. After examining agreement amongst group members on who they believed the group leader was, 22 total groups of the 43 had 66% agreement on who the leader was. These 22 groups were used for further data aggregation. From these 22 groups, individual member ratings for 5 variables, namely authentic pride, hubristic pride, leader satisfaction, team cohesion/satisfaction, and task satisfaction, were then aggregated across groups to arrive at 110 group-level variables. We then checked normality assumptions for these data.

#### 2.4.2. Normality Violations

(a) Individual Level Violation Correction:

Conducting analysis at either individual or group level, statistical assumptions were checked. For individual analyses, hubristic pride and task satisfaction ratings were highly skewed and kurtotic. Additionally, measures of leadership satisfaction, and team cohesion/satisfaction were negatively skewed. This could be explained by the task inherently being enjoyable and based on small team size. To correct for positive skewness and platykurtosis, a logarithmic transformation [48] was conducted on the leadership satisfaction and task satisfaction variables which corrected the skew and kurtosis. The sole exception was the average hubristic pride rating, which remained positively skewed even after transformation. These transformed variables (including hubristic pride) were used in all reported analyses.

(b) Group Level Violation Correction:

As in the individual level analysis, assumptions of skewness and kurtosis were both violated in many of the measures for the group level data. Again, these violations were addressed by logarithmically transforming the scores to correct for skewness and kurtosis violations [49]. Group-level hubristic leader ratings were positively skewed, while leadership satisfaction, task satisfaction, and team cohesion/satisfaction were all negatively skewed. After logarithmically transforming the data to correct for skewness, only hubristic pride remained positively skewed. This may have been a result of group members collectively believing that their agreed-on leaders did not generally display hubristic behaviors or that groups who agreed on their leader tended to be less hubristic generally.

Regarding kurtosis, only the measures of task satisfaction and hubristic pride ratings were both platykurtotic. However, after logarithmic transformation, kurtosis was corrected on both measures. The transformed variables were used in all further group-level analyses. We discuss the skew issue in our discussion section.

## 3. Results

### 3.1. Individual Level Results

Table 1 provides descriptive statistics and correlations of the major research variables. Age and gender had significant relationships with task satisfaction (older, and females were more satisfied with the task). Females were also more likely to be satisfied with the leader and group, as well as be rated as displaying more authentic pride. Regarding hypothesized relationships, in general, we see ratings of leaders’ authentic pride significantly positively associated with positive outcomes: task satisfaction, leader satisfaction, and team cohesion/satisfaction. However, there were no significant relationships with hubristic pride. When run separately, team cohesion was highly significantly associated with authentic pride: *r* = 0.46, *p* < 0.001; team satisfaction was also highly significantly associated with authentic pride: *r* = 0.58, *p* < 0.001. For hubristic pride, the correlation was significant for team cohesion, *r* = −0.13, *p* < 0.05 and marginally significant for team satisfaction, *r* = −0.05, *p* = *n.s*.

As noted above, we further examined leader perceived authentic and leader perceived hubristic pride as predictors of all three outcome variables (leader satisfaction, team cohesion/satisfaction, task satisfaction), with and without controlling for any significant (from correlations) demographic variables (per recent experts’ recommendations: Becker et al. [49] and Bernerth & Aguinis [50]) in the first block, and then perceived leader displayed authentic or hubristic pride in a second block. If the control variables were not significant in the regression analysis, we removed them to preserve statistical power, and reran the regression. If the control variables were significant, or approaching significance, we kept them in the model and report them and their interpretation. This was done because we wanted a clear test of displayed pride’s influence on our dependent variables, above and beyond any possible demographic contributions.

Leader perceived authentic pride was significantly positively related to all three outcome variables, above and beyond demographic variables. Perceived leader authentic displayed pride most strongly predicted team cohesion/satisfaction: Δ*R*^2^ = 0.36, *p* < 0.001, *b* = 0.60, *p* < 0.001; it had a similarly strong relationship with leader satisfaction: Δ*R*^2^ = 0.36, *p* < 0.001, *b* = 0.62, *p* < 0.001; and least for task satisfaction (but also significant): Δ*R*^2^ = 0.05, *b* = 0.23, *p* < 0.01. Leader hubristic pride was not significantly related to any of the dependent variables (See Table 2). Thus, hypotheses 1 a–c regarding authentic pride on the three outcomes were all supported; and 2 a–c regarding hubristic pride on outcomes was not.

To examine moderating effects of task, we ran Hayes Process Macros for SPSS (Model 1, one moderator), for each of the three DV’s, separately for authentic and hubristic pride. One model each for authentic and hubristic pride was significant, and for different DV’s.

For authentic pride, the model was significant for team cohesion/satisfaction, *R*^2^ = 0.39, *p* < 0.0001, with the main effect of authentic pride and the interaction terms being significant (but not task type, as a main effect). In the creative condition, authentic pride had a stronger effect (*b* = 0.47, *p* < 0.001) on team cohesion/satisfaction than in the detailed condition (*b* = 0.28, *p* < 0.001). Figure 1 shows the significant interaction, which is in the proposed direction and shows that average authentic pride in general is better, regardless of task type, but especially for creative tasks. This therefore supports Hypothesis 3c, for team cohesion/satisfaction.

For hubristic pride, the model was significant for task satisfaction, *R*^2^ = 0.05, *p* < 0.05, with the main effect of hubristic pride, task type, and the interaction terms being significant (or marginally so, for hubristic pride as a main effect). Participants were overall more satisfied with the creative task (*b* = −0.33; *p* < 0.05), and less satisfied with higher displayed hubristic pride. In the creative condition, hubristic pride had a negative effect (*b* = −0.18) on task satisfaction compared to the detailed condition where it had a positive effect (*b* = 0.18). This significant interaction with hubristic pride was not in the hypothesized direction. Essentially, Figure 1 shows that, for creative tasks, participants preferred (were more satisfied with the task) when the leader displayed low hubristic pride, but for detailed task, participants preferred (were more satisfied with the task) when the leader displayed high hubristic pride. Thus, hypotheses 4a–c were not supported.

### 3.2. Group Level Results

Following individual to group level data aggregation, we tested hypotheses 1 at the group level via correlations (see Table 3) regression analyses (see Table 4). In examining hypothesis 1, two of the three outcome variables aligned with what was predicted. Task satisfaction did not have a significant relationship with authentic pride at a group level, *r*(21) = 0.12, *n.s*. Leader authentic pride was significantly positively related to team satisfaction/cohesion (*r*(21) = 0.67, *p* < 0.01) and leadership satisfaction (*r*(21) = 0.77, *p* < 0.01). For hypothesis 2 only leadership satisfaction was negatively significantly related to hubristic pride (*r*(21) = −0.57, *p* < 0.01). The other outcome variables of team satisfaction/cohesion (*r*(21) = −0.33, *n.s*.) and task satisfaction (*r*(21) = 0.25, *n.s.*) were not significantly related.

Again, a moderation analysis was run as for the individual data to test hypotheses 3 and 4. None of the interactions were significant and thus neither hypotheses 3a–c nor 4a–c were supported at the group level. While the sample size for the group level hypotheses was 22, we find that the results are still worth reporting and show the regression results in Table 4 (without the non-significant interaction/moderators). These regression results completely mirrored the correlation results reported above. The results when using individual measure of cohesion and satisfaction differed as follows: Authentic Pride with cohesion separate, *r* = 0.48 *; with group satisfaction separate, *r* = 0.67 **; Hubristic Pride with cohesion separate, *r* = −0.37, *p* < 0.10; with group satisfaction separate, *r* = −0.16.

## 4. Discussion

We examined group members’ perceptions and reactions regarding their perceived leaders’ expressions of authentic and hubristic pride. Reviews of workplace affect have called for more attention to findings regarding discrete emotions (e.g., Gooty et al. [51]), of which pride is certainly one. As well, pride has received less attention than other, more common discrete emotions such as sadness and anger [7]. Further, we partially answered the “call” by Schaumberg and Tracy [7] (p. 16): “Whether or under which circumstances authentic pride and hubristic pride related to leadership effectiveness remains a more open question.”

As predicted, we found positive relationships for leader expressions of authentic pride, and negative (or no significant) effects for leader expressions of hubristic pride, on the outcomes of leader satisfaction, group cohesion/satisfaction, and task satisfaction. Moreover, we found an intriguing interaction/moderation effect for task type. On a detailed task, participants preferred a leader expressing hubristic pride, but for a creative task, they preferred leaders expressing more authentic pride.

Some unique aspects of our study are worth a brief discussion. For example, interestingly, when individual ratings were aggregated to the group level, 14 of the 22 groups (63.64%) were in the creative condition. This might reflect how for creative tasks, less concern is placed on who is leading, because individuals may view one single person as less necessary for creative output, than for a detailed one; after all, it is common for people to assume that groups are better for creative tasks [51].

We noted the skew for hubristic pride at the beginning of our results section, which was not completely corrected statistically after transformation. We should also note that overall levels of hubristic pride were quite low, as *M* = 0.31 (.24 for group level analysis) (*SD* = 0.56; 0.47, group) on a 0–4 scale. Yet, such restriction of range would make effects statistically more difficult to find (thus why we likely found fewer significant results for hubristic pride), which suggests to us the overall importance of the significant ones that we did find. The fact that even such small amounts of displayed hubristic pride were differentially related to outcomes is compelling and likely underscores their effects in actual work contexts where people probably have a much wider range of hubristic pride expressions.

Our results further substantiated the current nomological network of pride with the generally negative effects and perceptions of hubristic pride, and the generally positive effects of authentic pride [2,13,15,18,19,20,21], such that perceived expressions of authentic pride were positively associated with leader, task and team satisfaction and cohesion. Hubristic pride had no significant relationships (individual level analyses), and one strong significant negative relationship with leader satisfaction at the group level. This finding is particularly compelling given that at the group level, there were different sources of evaluation of the leader; as well, there was restriction of range, and a small sample size both making a significant relationship harder to detect statistically. This suggests that groups react quite strongly and negatively to even a hint of hubris displayed by their leaders.

Finally, there was one situation where displayed hubristic pride of leader was associated positively with outcomes: Figure 1 shows a direct relationship between hubristic pride rating and task satisfaction for the detailed task, such that at the individual level when hubristic pride ratings of the leader were high, task satisfaction was also high, for a detailed task. Contrary to this relationship, when the task was creative, task satisfaction held an inverse relationship with hubristic pride ratings.

One possible explanation for this effect is that, for challenging and/or tedious tasks, people might prefer a leader who acts extremely confidently (even if they find the person annoying!), because it boosts folks’ confidence that they will be successful: when people project confidence in themselves, others feel more confident in them [52,53]. In this context, we note that average task satisfaction was also rated as (marginally) significantly higher in the creative condition (*M* = 1.59, *SD* = 0.27) than in the detailed condition (*M* = 1.48, *SD* = 0.32), *t*(94) = 1.79, *p* = 0.08. Thus, essentially, we found hubristic pride expressions were more predictive of satisfaction on a less satisfying task. Perhaps there are yet undiscovered task contextual effects in operation that follow different mores for highly hubristic leaders.

For example, an earlier study examined leadership styles (relationship, task, and differing combinations of each) with subordinate dogmatism, interacting with four different types of tasks of differing levels of ambiguity and difficulty [54]. Although a detailed explanation of the significant three-way interaction these authors found is beyond the scope of our purpose here, a quote from these researchers illustrates our findings and conclusions regarding task moderation: “The match between a leader’s style and his subordinate’s personality is more important when the task is difficult and ambiguous than when it is easy and structured “(p. 63).

Also speaking to the pattern we found for task moderation are the results from a study conducted by Madlock [55] examined whether task (i.e., providing structure of the work to followers) or relational (i.e., providing nurturing support to followers) leadership style would be more predictive of job satisfaction He discovered significant and comparable correlations between task (*r* = 0.24) and relational (*r* = 0.29) leadership styles when correlated with job satisfaction within a variety of jobs. Our study expands on this finding by revealing leaders who display hubristic attributes may be more adept at providing task related leadership than authentic leader. In addition, depending on the nature of the job, one leadership style—task leadership—may be more appropriate in a job that is more detail-oriented than a relational leadership style. Although the constructs of relational and task leadership styles are different than displayed authentic and hubristic pride in group leaders, our finding sheds light on an additional leadership attribute—displayed authentic or hubristic pride—that may be more appropriate in a given work situation. Specifically, hubristically proud leaders may be more appropriate in work situation where the task at hand requires more detailed guidance, while authentically proud leaders may be more appropriate in creative work conditions that are less structured.

Generally, though, leaders should work to generate cultures where expressions of authentic pride (and feelings, if possible) are modeled and respected. Felt authentic pride can increase effort on tedious tasks [13]; and can be motivational [9]. It would, therefore, appear that feeling and expressing pride in one’s (legitimate) accomplishments might trigger a whole host of positive outcomes. This echoes what we mentioned earlier from Bodolica and Spraggon [3] who suggested leaders develop employees to feel comfortable in both feeling authentic pride in their accomplishments and then subsequently expressing it. As well, perhaps seeing people feel comfortable expressing authentic pride and that being valued, might trigger a contextual and cultural spiraling of generally valuing authentic emotional expression. As calls for authenticity in life and at work increase, and research continues to show positive effects for supporting such (and negative effects for felt inability to be authentic; see Sutton [56] for a recent meta-analysis), feeling pride in one’s accomplishments and being comfortable to express it would seem quite timely as advice for managers, leaders, and organizations.

### 4.1. Practical Implications

Our findings are important for several practical reasons. Our work suggests that leaders need to be aware of their expression of emotion, and of being perceived as conceited, perhaps when they thought they were expressing confidence or were “just joking.” The onerous (potential and actual) impact of leader narcissism in organizations has recently been reviewed [57] and has been illustrated, e.g., by Blair et al. [58]. Due to hubristic pride’s known relationship with narcissism [6], our results add to the collective narcissism/hubris-negative outcomes paradigm.

Our context (group task) and incentive we offered (Jimmy John’s gift card, which students appeared quite excited about) would likely enhance the above effect further due to participants wanting to do well and needing to rely on others to do so. The literature on narcissism and the positive relationship between narcissism and hubristic pride discussed above is also of relevance here and to our predictions. Although it has been found that in the short-term narcissistic (self-enhancer) people may be liked, in the long-term, they became less liked [59]. Perhaps this short-term attraction is due to perceived competence and self-esteem, signaling that the person might be able to help achieve a group goal. Alternatively, a person expressing hubris might detract from perceptions that they could help in a creative task, as it requires different skills.

### 4.2. Limitations and Future Research

The primary limitations of our study include commonly touted ones pertaining to laboratory environments, and cross-sectional studies. Our artificial environment (short-term; non-important task) and student sample make external and ecological validity challenging. It is a stretch to suggest definitively that our findings apply to “real” leaders of “real” groups working on a “real” task in the “real” world. Yet upon reflection, we find no reason to suggest that others’ perceptions of leader displayed authentic or hubristic pride should be different in most any context. Authenticity is appreciated [56] and hubris is disliked [59,60]. However, similar research in an applied context would be welcome, particularly regarding tasks of different levels of importance.

Although cross sectional, our study did not suffer from standard criticism regarding using only self-report measurement: the fact that we had people rate their own outcome variables, but the leaders’ pride expressions, renders the typical worry regarding common method variance somewhat moot. Also, the different patterns for the different dependent variables suggest that there was a distinction among concepts occurring for participants, rather than them simply relying on a macro level positive or negative frame. Further, given that we found consistent results when we aggregated to the group level as for our analyses at the individual level, helps support that these results are not simply artifacts. Recall our group level data were only for those where the group agreed as to whom the leader was, so it could not have been the case that all of those people were thinking about themselves when assessing pride expressions (but they were all reporting their own satisfaction). We are thus confident that expressions of authentic pride have fairly consistent positive effects on followers, and expressions of hubristic pride have (less) consistent negative effects.

In addition, our non-experimental design makes causal statements untenable. Indeed, as described in the methods section, we had originally attempted to test our ideas with an experimental design (using a trained leader (confederate) acting/expressing authentic versus hubristic pride) which would have made the step in assessing agreement on who people thought was the leader moot (as there would have been one, clear leader). Perhaps future research could figure ways to manipulate expressions of authentic and hubristic pride, such as through using a videotaped interaction with actors, trained generally to express authentic or hubristic pride several times.

Further, both our collapsing group cohesion with group satisfaction, and our measure of group cohesion could be considered weaknesses. First, due to highly related constructs/measures (both statistically and conceptually), we chose to analyze the relationship with the collapsed variable, for parsimony (and to save statistical power for the group level analysis which was already underpowered). We note that after running all analyses both with the collapsed variables together and separate, the results did not appreciably differ. As well, group cohesion is often defined in a more nuanced fashion, [34]—with two dimensions of task cohesion and interpersonal cohesion. This particular author showed that they both had differing influences on team satisfaction. Researchers who seek to extend and build upon our findings might delve into the measurement of these related constructs, and attempt to tease out the nomological network of relationships among them.

We also note that, at the group level, the analysis was severely underpowered. The sample size was only 22, falling below the threshold limit power of results for group level results. For the number of variables (two IVs and four DVs), sample size ideally would be no less than 100. Some of the correlations in the group level analysis were, indeed, moderate in size, but not significant, thus underscoring this statistical underpowering. As mentioned, we thus find the significant results that we found at the group level particularly compelling.

Finally, although we had no strong ideas or hypotheses originally regarding performance on the Lego task (it was a peripheral interest; our core interest was the various satisfaction attitudes), we were intending to examine (in an exploratory fashion) if authentic leader pride would be positively related to performance (both detailed and creative), and hubristic would be inversely related. The lost data was thus quite disappointing, and we would encourage future research to examine task performance as a dependent variable, for various types of tasks.

Future research should examine explanatory mechanisms for the operation of authentic and hubristic pride on followers: for example, is it due to global positive attributions of the leader when they express more authentic pride? Is it the esteem-enhancing effect on followers that authentic pride signals (in contrast to the esteem-reducing signals from hubristic pride)?

Future research also should seek to disentangle this narcissism (and by default, projected hubris)–love/hate relationship. Is it the case that while we may not “like” narcissism and hubris, there might be times where we want such a leader…when we perceive their confidence and strength would be helpful? Recent research examining outcomes from the last US Presidential election [59] found that narcissism, and its relationship with outcomes (voter choice, in this study), might differ according to the person, and the consequent attributions made regarding that person (attributed charisma, in this study). In some cases, for example, perceptions of charisma (and resultant confidence and power attributions) can somewhat buffer narcissism. Moreover, we know that narcissistic people tend to wear out their welcome over time [59], and thus we could expect that leaders expressing hubris would also not be as well-liked long term, when the particular task at hand is less of a signal. Future research could examine trajectories of satisfaction with leaders of those with differing levels of narcissism and displayed hubris, to substantiate these suggestions.

In this context, future research could also look into whether or not authentic or hubristic pride in leaders could predict their emergence and ratings for creative versus detailed tasks, as well as other types of tasks such as decision-making versus brainstorming. Finally, although we conceptually tied displayed hubristic pride to narcissistic personality, we should reinforce that we only examined displayed discrete emotions. Future research might establish ways to examine leaders with more trait authentic pride versus those with more trait hubristic pride, (likely) assuming outcomes on followers would follow a mediational path through discrete authentic and hubristic pride expressions.

## 5. Conclusions

In conclusion, our study found that leader expressed authentic pride was significantly related to individual satisfaction with the leader, individual satisfaction with the task, and individual ratings of (collapsed) group cohesion/satisfaction. At the group level, aggregated perceptions of leader authentic pride were positively related to satisfaction with the leader as well as group perceived cohesion and satisfaction. Additionally, at the group level, perceived leader hubristic pride was negatively related to satisfaction with the leader. In analyzing moderating effects at the individual level, we also discovered two interesting findings regarding the context of the task. Individuals who were in the creative condition experienced more team cohesion and satisfaction by having an authentically proud leader than in the detailed task condition. In the detailed task condition, group members who rated their leaders as being authentically proud did not experience feelings of team cohesion and satisfaction as intensely as those in the creative condition. However, there was still a positive impact on team cohesion and satisfaction of having an authentically proud leader. Additionally, when the task was detailed in nature, task satisfaction was higher when hubristic ratings were higher. Taken together, the task a group undertakes impacts the relationship between perceived authentic and hubristic pride and two different outcome variables, namely team cohesion and task satisfaction. These findings generally support the proposition that authentic pride leads to positive outcomes in group tasks both at the individual and collective level with nuance regarding the context of the task in group tasks. We hope our research provides new insights for organizational scholars and practitioners alike.

## Figures and Tables

**Figure 1 ijerph-17-07146-f001:**
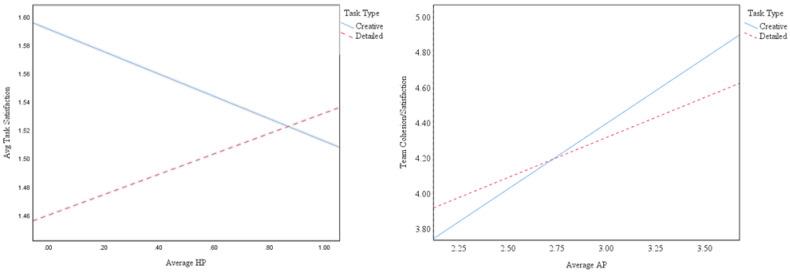
Moderation of task type on authentic pride’s relationship with team cohesion/satisfaction (Left) and moderation of task type on hubristic pride’s relationship with task satisfaction (Right).

**Table 1 ijerph-17-07146-t001:** Descriptives and Correlation table between research variables (*N* = 179).

Variables	M	SD	1	2	3	4	5	6	7
1. Age	19.49	1.92							
2. Gender ^a^			0.15 *						
3. Leader Auth. Pride ^b^	2.94	0.62	0.08	−0.18 **	0.88				
4. Leader Hubristic Pride ^b^	0.31	0.56	−0.04	0.02	−0.06	0.9			
5. Leadership Satisfaction ^b^	2.98	0.77	0.05	−0.11	0.62 **	−0.06	0.84		
6. Team Cohesion/Satisfaction ^c^	4.31	0.6	0	−0.13 *	0.60 **	−0.11	0.72 **	0.87	
7. Task Satisfaction ^d^	1.52	0.29	0.16 *	−0.15 *	0.26 **	0.01	0.28 **	0.39 **	0.7

Note: Sample sizes change due to missing data. * *p* < 0.05 (one-tailed). ** *p* < 0.01 (one-tailed). ^a^ 1—male; 0—female; ^b^ Authentic and Hubristic Leadership, leader, and team satisfaction average scores could all hypothetically range from 0 (not at all) to 4 (extremely); ^c^ Team cohesion/satisfaction average could hypothetically range from 1–7.; ^d^ Task satisfaction average could hypothetically range from 0–2.

**Table 2 ijerph-17-07146-t002:** Individual level regressions with perceived leader authentic and hubristic pride as predictors.

Independent Variable	Dependent Variable	B	SE B	Beta	Sig
Perceived Leader Authentic Pride Rating	1. Leader Satisfaction *	0.61	0.06	0.62	0.001
	2. Team Cohesion/Satisfaction *	0.53	0.05	0.60	0.001
	3. Task Satisfaction **	0.22	0.07	0.23	0.002
Perceived Leader Hubristic Pride Rating	4. Leader Satisfaction	−0.07	−0.01	−0.06	0.407
	5. Team Cohesion/Satisfaction	−0.11	0.08	−0.11	0.165
	6. Task Satisfaction	0.01	0.09	0.01	0.871

Note: * Controlled for gender; ** Controlled for age and gender. Gender and age were controlled for only when they were found to significantly impact the regression model. All regressions were conducted using a hierarchical regression.

**Table 3 ijerph-17-07146-t003:** Correlation table of group aggregated leader expressed pride and outcome variables (*N* = 22).

Variables	Mean	Standard Deviation	1	2	3	4
1. Leader Hubristic Pride	0.24	0.21				
2. Leader Authentic Pride	3.01	0.26	−0.52 **			
3. Leadership Satisfaction	3.1	0.33	−0.57 **	0.77 **		
4. Team Cohesion/Satisfaction	4.34	0.31	−0.33	0.67 **	0.70 **	
5. Task Satisfaction	1.55	0.16	0.25	0.12	0.08	0.47 *

Note: Pearson’s one tailed test; * *p* < 0.05 (one-tailed); ** *p* < 0.01 (one-tailed). Authentic and Hubristic Leadership, leader, and team satisfaction average scores could all hypothetically range from 0 (not at all)–4 (extremely); Task satisfaction average could hypothetically range from 0–2; Team cohesion/satisfaction average could hypothetically range from 1–7.

**Table 4 ijerph-17-07146-t004:** Group level regression results with Authentic and Hubristic Pride as predictors.

Independent Variable	Dependent Variable	B	SE B	Beta	Sig
Perceived Leader Authentic Pride Rating	1. Leader Satisfaction	0.71	0.15	0.73	0.000
	2. Team Cohesion/Satisfaction	0.66	0.17	0.66	0.001
	3. Task Satisfaction	−0.03	0.31	−0.02	0.921
Perceived Leader Hubristic Pride Rating	4. Leader Satisfaction	−0.47	0.19	−0.49	0.021
	5. Team Cohesion/Satisfaction	−0.13	0.22	−0.13	0.570
	6. Task Satisfaction	0.57	0.28	0.42	0.053

Note: *N* = 22.

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
