# Peer review of "Ain’t Too Proud to Beg! Effects of Leader’s Use of Pride on Groups"

_ijerph, 2020, doi:10.3390/ijerph17197146_

Round 1
Reviewer 1 Report
Research is important and its results very interesting, however its presentation can be improved in the article that is submitting to the Journal. Here I suggest some changes and additions that may be useful:
1. Introduction
The introduction does not clear to highlight the importance of study. Should be more detailed and rewrote-up the whole of the introduction section, what is the main problem, objective, and question of the study. Page 1
2. Review
1.1.5 Task Satisfaction
This section is not clear because is too short explained about task satisfaction. Page 4
3. Methods
Should be cleared how the authors' selection the sample techniques and sampling process.
What interview/questionnaire form was used?
How you measure the variable in the statistical analysis? Because the used of difference in measuring scale.
This is not clear how did you used the statistical for analytical process.
There is no analysis section. The author should be having this section because invalid analysis process. If analysed by researchers, the analysis of the study is incorrect.
4. Results
The author never goes back to test the hypothesis, why some variables are insignificant and some variables are significant. Should be explained what values are mean and reflected to the study findings.
5. Discussion
Should be updated and in-depth discussed with main findings and theoretical contributions.
6. Conclusion
The conclusion should be enriched as follows: summarizing the core research findings, recommendation to confront the finding with the results.
Author Response
Reviewer 1 comments addressed.
- “The introduction does not clear to highlight the importance of study. Should be more detailed and rewrote-up the whole of the introduction section, what is the main problem, objective, and question of the study. Page 1”
Response: We added a clarifying paragraph with a summary of our contributions and tried to elucidate them. We disagree we need to “rewrote-up” the whole of the introduction and feel we reviewed the literature quite well, in general. We feel similarly regarding this reviewer’s comment regarding our (poor) English: We simply disagree.
- “1.1.5 Task Satisfaction This section is not clear because is too short explained about task satisfaction”
Response: We added more information about job satisfaction and leadership in general, and a expanded a bit more in this section in general.
- “Should be cleared how the authors' selection the sample techniques and sampling process.
Response: We added more information about the introductory psychology participant pool, how we recruited students.
- What interview/questionnaire form was used?
Response: We were not sure as to what this meant as all our measures were explicitly discussed in the manuscript.
- How you measure the variable in the statistical analysis? Because the used of difference in measuring scale. This is not clear how did you used the statistical for analytical process. There is no analysis section. The author should be having this section because invalid analysis process. If analysed by researchers, the analysis of the study is incorrect”
Response: this was too hard to decipher meaning here, but our original paper certainly has a measures section and also analyses, but we did add a data analysis section, recommended by both reviewers 1 and 2.
- “The author never goes back to test the hypothesis, why some variables are insignificant and some variables are significant. Should be explained what values are mean and reflected to the study findings.”
Response: This is an inaccurate statement. We tested and reported each in the results section.
- “Should be updated and in-depth discussed with main findings and theoretical contributions.”
Response: We added more in depth discussion of some things, as well as including two helpful articles for interpreting our interaction that R2 suggested.
- “The conclusion should be enriched as follows: summarizing the core research findings, recommendation to confront the finding with the results.”
Response: The latter part of this statement is too difficult to interpret; we did add a final conclusion paragraph and we already had a limitations and future research section, if that is what the latter part of the sentence meant.
Reviewer 2 Report
This is an interesting well-argued article that would contribute to the existing knowledge about leadership and its effects on followers and tasks. However, it requires some revisions before it can be published.
Introduction
The bibliographic review is extensive and up-to-date. Nevertheless, sometimes the argumentation is somewhat repetitive (i.e., p.4, lines 182-185; p.5, lines 207-209).
On the other hand, although the authors strive to present their explanations in a systematic way, they should consider condensing all hypotheses at the end of the introduction and omitting premature references to them (ie. p 3, lines 106-111; p.4, lines 186-191).
Just before the hypotheses, the aim of the study and the differential contribution that it represents should be explicit for the reader, given the antecedents presented.
The allusion in the introduction to the psychometric justification of combining group cohesion and group satisfaction is unnecessary. A theoretical justification should be given here. Personally I consider this justification difficult to achieve, since most of the bibliography considers them as two independent variables, although certainly related, as the authors point out. For example:
Carmen Picazo, Nuria Gamero, Ana Zornoza & Jose M. Peiró (2015) Testing relations between group cohesion and satisfaction in project teams: A cross-level and cross-lagged approach , European Journal of Work and Organizational Psychology, 24:2, 297-307, DOI: 10.1080/1359432X.2014.894979
Materials and Methods
The first measure they describe is not contemplated in the hypotheses, nor in the antecedents. Although it would certainly be very interesting to know if, for example, authoritarian leadership positively correlates with hubristic pride and in turn both generate greater satisfaction in detailed tasks, I consider that this description of the instrument could be omitted, since instead of contributing to the results it distracts the reader.
The statistical justification they use to link satisfaction and group cohesion is insufficient. Statistics can sometimes provide us with capricious results, which, if not accompanied by theoretical fundamentals, are meaningless. If the authors do not find a theoretical way to justify joining these two variables, they should consider showing them separately.
In the procedure they refer to the delivery of vouchers from "Jimmy John's", please clarify that it is a local restaurant.
Perhaps, the most relevant aspect that should be introduced in this manuscript is the heading “2.4. Data Analysis ”. Although in the results we can see the operations carried out, in this section they must explain what they have done (correlations, regression analysis ...), including the ways in which they tested the statistical assumptions (p.7, line 321-329), the criteria to evaluate effect sizes, and so on.
Apparently, the statistical operations used are adequate to test the hypotheses.
Results
There are many explanations about the operations carried out mixed with the results, if the authors create the epigraph “2.4. Data Analysis ”, the results would gain in clarity (p.8, lines 344-353; p.8, lines 363-365). This makes the results difficult to read and understand.
A table that summarizes the regression analysis models performed would be appreciated.
The difference and justification between individual and group analyzes should also be better explained in the new section 2.4.
Discussion and Conclusions
They are extensive and include the results obtained. One of the most interesting aspects is the possible explanation of how in a detailed task, hubristic pride had a good impact in satisfaction. This appears treated in two different paragraphs (p. 12, lines 459-461; lines 482-494), perhaps they could unify it in a single paragraph. They could even enrich it a bit more by referring to leadership studies such as:
Weed S.E., Mitchell T.R., Moffitt W. (1981) Leadership Style, Subordinate Personality and Task Type as Predictors of Performance and Satisfaction with Supervision. In: Gruneberg M.M., Oborne D.J. (eds) Psychology and Industrial Productivity. Palgrave Macmillan, London. https://doi.org/10.1007/978-1-349-04809-0_9
Madlock, P. E. (2008). The Link Between Leadership Style, Communicator Competence, and Employee Satisfaction. The Journal of Business Communication (1973), 45 (1), 61–78. https://doi.org/10.1177/0021943607309351
The authors are correct in clarifying that this is an exploratory study (p.13, line 532), although this should also be noted in the method section.
The Funding, Acknowledgments and Conflicts of Interest sections are unresolved.
Author Response
- “On the other hand, although the authors strive to present their explanations in a systematic way, they should consider condensing all hypotheses at the end of the introduction and omitting premature references to them (ie. p 3, lines 106-111; p.4, lines 186-191).”
Response: Thank you for this suggestion; We revised the ms accordingly.
- “…authors strive to present their explanations in a systematic way, they should consider condensing all hypotheses at the end of the introduction and omitting premature references to them (ie. p 3, lines 106-111; p.4, lines 186-191).”
Response: ibid
- “The allusion in the introduction to the psychometric justification of combining group cohesion and group satisfaction is unnecessary. A theoretical justification should be given here. Personally I consider this justification difficult to achieve, since most of the bibliography considers them as two independent variables, although certainly related, as the authors point out.
Response: Thanks for this [correct] call/comment/suggestion. We expanded much around this issue in several places. First, in defining the two, we added literature which clarifies that the two constructs are, indeed, distinct. Thank you for the helpful reference, it is included in this section (as well as several others we found) as well as in the discussion mentioning the collapsing of these two as a limitation. In poring over definitions, it became quite clear that what we found statistically (high degree of overlap) which cued us to collapse, was also represented in the high degree of overlap in some definitions. As well, we are more transparent in the revised ms that we actually intended to keep them separate, but for parsimony (and statistical power), reported the collapsed variables. We add in footnotes noting appreciable differences statistically when the variables are separate. None were substantive, but we are again attempting to be transparent with you, the reviewer, and with our readers. Finally, we add a section to the discussion calling attention to this. If this reviewer still strongly prefers we report everything with the variables kept separate, we would be willing to do so.
- “Just before the hypotheses, the aim of the study and the differential contribution that it represents should be explicit for the reader, given the antecedents presented”
Response: we added a short paragraph as to such.
- “sometimes the argumentation is somewhat repetitive (i.e., p.4, lines 182-185; p.5, lines 207-209)”
Response – We removed the repetitions. (E.g. deleted lines 207-209, etc.)
- The first measure they describe is not contemplated in the hypotheses, nor in the antecedents. Although it would certainly be very interesting to know if, for example, authoritarian leadership positively correlates with hubristic pride and in turn both generate greater satisfaction in detailed tasks, I consider that this description of the instrument could be omitted, since instead of contributing to the results it distracts the reader.”
Response: We cut this and put it in a footnote for transparency (and noted the low reliabilities of these scales rendering them unusable).
- “The statistical justification they use to link satisfaction and group cohesion is insufficient. Statistics can sometimes provide us with capricious results, which, if not accompanied by theoretical fundamentals, are meaningless. If the authors do not find a theoretical way to justify joining these two variables, they should consider showing them separately.”
Response: See above
- “In the procedure they refer to the delivery of vouchers from "Jimmy John's", please clarify that it is a local restaurant.”
Response: Clarified.
- “Perhaps, the most relevant aspect that should be introduced in this manuscript is the heading “2.4. Data Analysis ”. Although in the results we can see the operations carried out, in this section they must explain what they have done (correlations, regression analysis ...), including the ways in which they tested the statistical assumptions (p.7, line 321-329), the criteria to evaluate effect sizes, and so on.”
The difference and justification between individual and group analyzes should also be better explained in the new section 2.4.
Response: We found this suggestion (also by reviewer 3) to be quite helpful and added a rather extensive section of Data Analysis which includes some material cut from other places and moved into this section, and some new material to help clarify what we did.
- “A table that summarizes the regression analysis models performed would be appreciated.”
Response: Thank you for this. Two regression tables added.
- Reviewer 2: “There are many explanations about the operations carried out mixed with the results, if the authors create the epigraph “2.4. Data Analysis”, the results would gain in clarity (p.8, lines 344-353; p.8, lines 363-365). This makes the results difficult to read and understand.”
Response: see above
- “They are extensive and include the results obtained. One of the most interesting aspects is the possible explanation of how in a detailed task, hubristic pride had a good impact in satisfaction. This appears treated in two different paragraphs (p. 12, lines 459-461; lines 482-494), perhaps they could unify it in a single paragraph. They could even enrich it a bit more by referring to leadership studies such as:
Weed S.E., Mitchell T.R., Moffitt W. (1981) Leadership Style, Subordinate Personality and Task Type as Predictors of Performance and Satisfaction with Supervision. In: Gruneberg M.M., Oborne D.J. (eds) Psychology and Industrial Productivity. Palgrave Macmillan, London. https://doi.org/10.1007/978-1-349-04809-0_9
Madlock, P. E. (2008). The Link Between Leadership Style, Communicator Competence, and Employee Satisfaction. The Journal of Business Communication (1973), 45 (1), 61–78. https://doi.org/10.1177/0021943607309351”
Response: This was exceptionally helpful to us; we also found the interaction to be the most interesting finding and we appreciate the references you suggested. We expanded on this quite a bit in the discussion section and incorporated information from those suggested references.
- “The authors are correct in clarifying that this is an exploratory study (p.13, line 532), although this should also be noted in the method section.”
Response: We were unsure about this comment; we tried to explain a bit more in the discussion what we meant with the word exploratory. However, we note here that our study was not exploratory, as we had clear, a priori hypotheses which we tested. The piece which would have been totally exploratory was the analysis of the [missing] performance data (as another DV) which we never had any solid predictions about. We hope we clarified this appropriately and addressed this reviewer’s concerns.
- “The Funding, Acknowledgments and Conflicts of Interest sections are unresolved.”
Response: Done.
Reviewer 3 Report
I will make my comments brief.
Line 276-279 argues against the appropriateness of the design. I agree.
The design path taken now includes potential factorial combinations of identified perceived leaders and associated perceived authentic or hubristic pride. For example, each individual in a group could identify a different leader. I'm trying to diagram all the paths for correlations in my head. WOW.
This exercise need to be carefully rethought. Thinking about that, I was surprised to not see more said about the results or loadings of the LSQ. Certainly that scale would inform your findings.
Having said that, I straddled between reject and major revision.
Author Response
- “The design path taken now includes potential factorial combinations of identified perceived leaders and associated perceived authentic or hubristic pride. For example, each individual in a group could identify a different leader. I'm trying to diagram all the paths for correlations in my head. WOW.”
Response: we tried to add clarifying sentences and paragraphs. So we made clearer (we hope) that individual level analysis was each person's perceptions of the leader, whereas group analysis was aggregated by only groups with enough agreement as to leader
- “This exercise needs to be carefully rethought. Thinking about that, I was surprised to not see more said about the results or loadings of the LSQ. Certainly, that scale would inform your findings.”
Response: We took it out per reviewer 2 and due to its low reliability rendering it unusable.
- “Line 276-279 argues against the appropriateness of the design. I agree.”
Response: We are unsure if this reviewer thought our currently presented design was inappropriate or if they thought what we attempted (and failed) was inappropriate. If the former, we respectfully disagree and argued for why this made sense to us, even though our initial plan was not tenable. If the latter, we are glad that you understand why the attempted experimental design simply would not work.
Round 2
Reviewer 2 Report
I believe that the authors have made a great effort to correct the manuscript. And that this effort deserves the publication of their work. Congratulations to its authors and the journal.